# Room temperature and low-field resonant enhancement of spin Seebeck effect in partially compensated magnets

R. Ramos [1*], T. Hioki [2], Y. Hashimoto[1], T. Kikkawa [1,2], P. Frey[3], A.J.E. Kreil [3], V.I. Vasyuchka[3], A.A. Serga [3], B. Hillebrands [3] & E. Saitoh[1,2,4,5,6]

Resonant enhancement of spin Seebeck effect (SSE) due to phonons was recently discovered in $Y_3Fe_5O_{12}$ (YIG). This effect is explained by hybridization between the magnon and phonon dispersions. However, this effect was observed at low temperatures and high magnetic fields, limiting the scope for applications. Here we report observation of phonon-resonant enhancement of SSE at room temperature and low magnetic field. We observe in $Lu_2BiFe_4GaO_{12}$ an enhancement 700% greater than that in a YIG film and at very low magnetic fields around $10^{-1}$ T, almost one order of magnitude lower than that of YIG. The result can be explained by the change in the magnon dispersion induced by magnetic compensation due to the presence of non-magnetic ion substitutions. Our study provides a way to tune the magnon response in a crystal by chemical doping, with potential applications for spintronic devices.

[1] WPI Advanced Institute for Materials Research, Tohoku University, Sendai 980-8577, Japan. [2] Institute for Materials Research, Tohoku University, Sendai 980-8577, Japan. [3] Fachbereich Physik and Landesforschungszentrum OPTIMAS, Technische Universität Kaiserslautern, 67663 Kaiserslautern, Germany. [4] Department of Applied Physics, The University of Tokyo, Tokyo 113-8656, Japan. [5] Center for Spintronics Research Network, Tohoku University, Sendai 980-8577, Japan. [6] Advanced Science Research Center, Japan Atomic Energy Agency, Tokai 319-1195, Japan. *email: ramosr@imr.tohoku.ac.jp

Heat is an ubiquitous and highly underexploited energy source, with about two-thirds of the used energy being lost as wasted heat[1], thus representing an opportunity for thermoelectric conversion devices[2]. Recently, a new thermoelectric conversion mechanism has emerged: the spin Seebeck effect (SSE)[3,4], driven by thermally induced magnetization dynamics in magnetic materials, which generates a spin current. The spin current is then injected into an adjacent metal layer, where it is converted into an electric current by the inverse spin Hall effect[5,6].

Lately, it has been shown that the magnetoelastic coupling can improve the conversion efficiency of SSE, as demonstrated by the observation of a resonant enhancement of the SSE voltage in YIG films[7,8]. The observed SSE enhancement has been explained by the magnon–phonon hybridization at the crossing points of the magnon and phonon dispersions due to momentum ($k$) and energy ($\hbar\omega$) matching; at certain magnetic-field values, the dispersions tangentially touch each other and the magnon–phonon hybridization effects are maximized (see Fig. 1a, b), resulting in peak structures due to the reinforcement of the magnon lifetime affected by the phonons. However, this effect has only been observed at low temperatures and high magnetic fields[7–14].

Engineering the magnon dispersion offers the possibility to tune the magnetic field at which the magnon–phonon hybridization effects are maximized, one possible approach is modification of the magnetic compensation of a ferrimagnet[15,16]. Intuitively, it can be expected that as the magnetic compensation of the system is introduced, the magnon dispersion gradually evolves from a parabolic $k$-dependence in the non-compensated state (similar to a ferromagnet) toward a linear dispersion when the system becomes fully compensated (i.e., zero net magnetization, antiferromagnetic case), schematically shown in Fig. 1c. Ideally, in the antiferromagnetic state in which spin-wave velocity is same as the sound velocity, the magnon–phonon coupling effects can be maximized even further as a consequence of potentially larger overlap between the magnon and phonon dispersions, however, this is yet to be experimentally observed.

In order to investigate the influence of increased magnetic compensation on the magnon–phonon coupling effects, we use a $Lu_2BiFe_4GaO_{12}$ (BiGa:LuIG) film as a model system and study the magnetic field and temperature dependences of the SSE. This system is a garnet ferrite, similar to YIG. It has two magnetic sublattices, where all the ions carrying spin angular momentum are $Fe^{3+}$. The ferrimagnetic order originates from the different site occupation between the two different magnetic sublattices, with the 3:2 ratio of tetrahedral (d) to octahedral (a) sites, resulting in nonzero magnetization. Substitution of Fe with Ga reinforces the magnetic compensation of the system due to the preferential occupation of the tetrahedral sites by the Ga ions[17] (Fig. 1d), resulting in the reduction of the magnetic moment and ordering temperature (see the Methods section and Supplementary Note 2 for more details on magnetic properties). The use of BiGa:LuIG film allows systematic evaluation of spin-wave dispersion using magneto-optical spectroscopy[18,19].

Here, we show that the doping affects the lifetime of magnons and the shape of their dispersion. This results in an increased magnitude of the resonant enhancement of SSE, which becomes observable at very low magnetic fields.

## Results

**Room temperature resonant enhancement of SSE.** We performed SSE measurements in the longitudinal SSE configuration, as schematically depicted in Fig. 2a. The magnetic-field dependence of the SSE voltage measured at 300 K is shown in Fig. 2b. We observed clear peaks in the voltage at magnetic-field values of $\mu_0 H_{TA} \sim 0.42$ T and $\mu_0 H_{LA} \sim 1.86$ T, as shown in the data blow-ups in Fig. 2c, d. These correspond to the hybridization of the

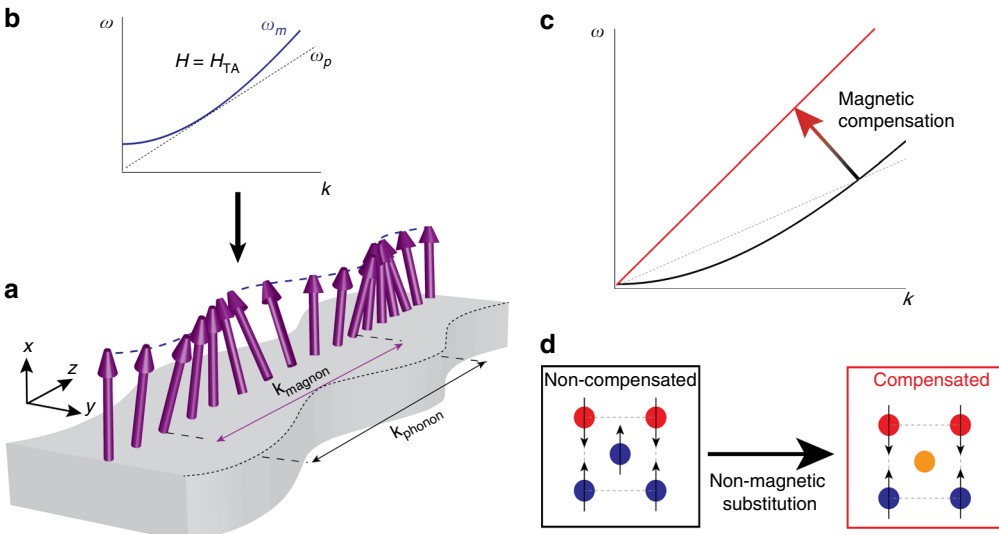

**Fig. 1** Lattice and spin waves in a magnetic material. **a** Schematic representation of a propagating phonon (lattice) and magnon (spin) excitation in a magnetic medium. Magnon–polarons are formed due to magnon–phonon hybridization when the magnon and phonon dispersions tangentially touch each other (i.e., they have coincident wavenumber $k_{magnon} = k_{phonon}$, and group velocity $\frac{\partial \omega}{\partial k}|_{magnon} = \frac{\partial \omega}{\partial k}|_{phonon}$), as schematically depicted by the dispersion relation shown in **b** representing the magnon (blue curve) and phonon dispersions (dashed line) under an applied magnetic field with magnitude $H_{TA}$, in this condition a resonant enhancement of the SSE can be observed. **c** Schematic representation of the effect of magnetic compensation on the magnon dispersion of a ferrimagnetic system: a parabolic dispersion is expected when the system is nonmagnetically compensated with nonzero magnetization (black curve), and as the magnetic compensation increases the magnon dispersion gradually evolves toward a linear dependence when the system reaches magnetic compensation (antiferromagnet). This can be experimentally achieved by introduction of nonmagnetic ion substitutions into the lattice as shown in panel **d**. Red and blue circles represent magnetic ions in two magnetic sublattices (a and d sites, respectively) with oppositely oriented spins and the orange circle represents the nonmagnetic ion substitution in d sites

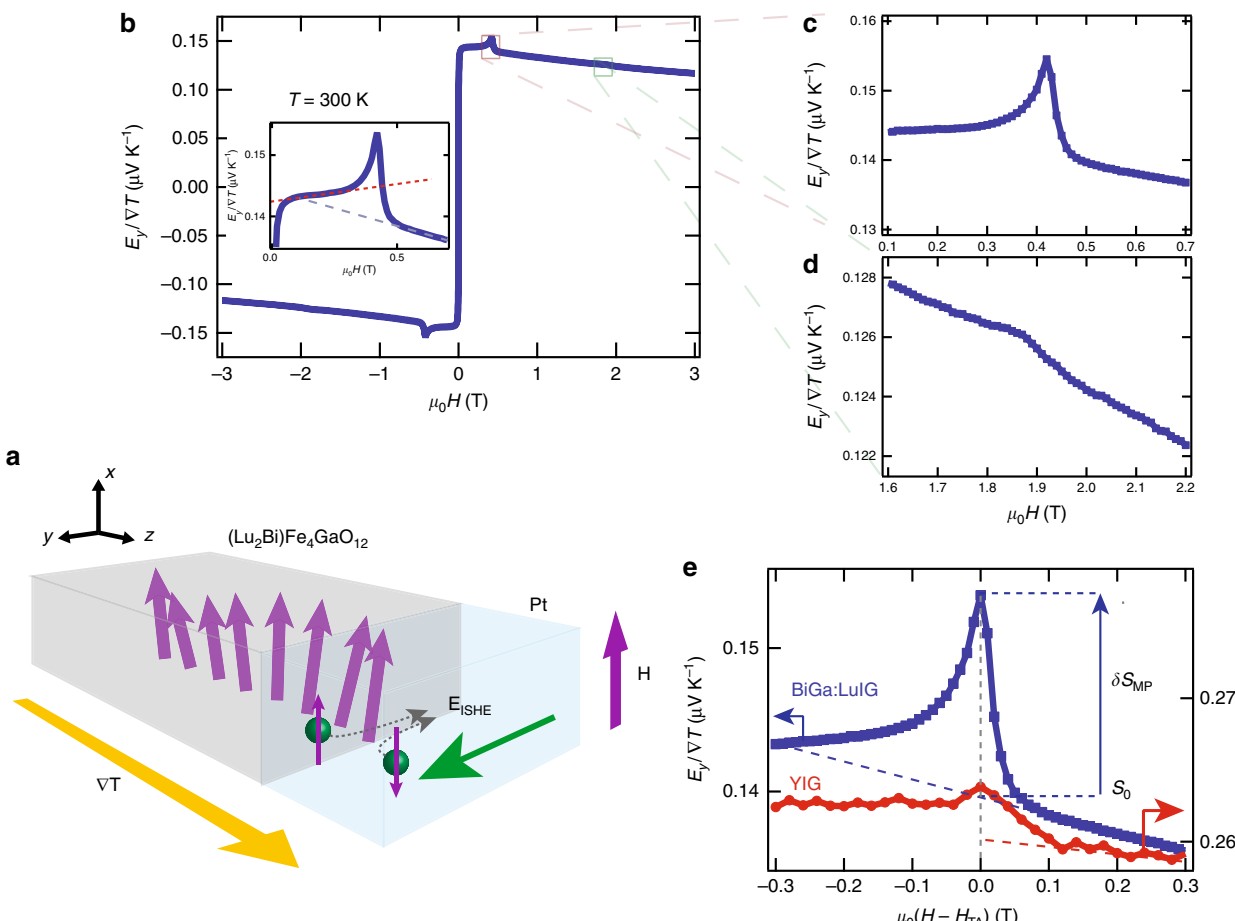

**Fig. 2** Spin Seebeck effect (SSE) measurement and magnon–polaron peaks in Ga-doped garnet system. **a** Schematic of the SSE and inverse spin Hall effect mechanism and measurement geometry. **b** Magnetic-field dependence of the SSE thermopower measured at $T = 300$ K in a $(Lu_2Bi)Fe_4GaO_{12}$ (BiGa:LuIG) film (normalized by the sample geometry: $E_y/\nabla T = S = (V/\Delta T)(L_z/L_y)$). Inset shows detail of the SSE in the $0 < \mu_0 H < 0.7$ T range, showing that the signal increase due to magnon–phonon hybridization is already present at very small fields. **c, d** Detail of the SSE signal in the vicinity of magnetic fields $\mu_0 H_{TA} = 0.42$ T and $\mu_0 H_{LA} = 1.86$ T, depicting the resonant enhancement of the SSE voltage, resulting from the magnon and phonon hybridization at the magnetic fields when the magnon dispersion just tangentially touches the transversal acoustic (TA) (**c**) and the longitudinal acoustic (LA) (**d**) phonon dispersions. The enhancement of the SSE at the peak positions is 10.21% and 0.4% for the TA and LA phonons, respectively (estimated as: $\delta S_{MP}/S_0$, where $S_0$ is the extrapolated background SSE coefficient at the peak position and $\delta S_{MP} = S(H_{MP}) - S_0$, with $S(H_{MP})$ the SSE coefficient at the MP (TA or LA) peak). **e** Comparison of the resonant enhancement of the SSE centered around the peak position ($H_{TA}$) for YIG and BiGa:LuIG at 300 K, $\delta S_{TA}/S_0 = 1.27\%$ for YIG

magnons with the transversal acoustic (TA) and longitudinal acoustic (LA) phonons, respectively. The observation of the clear enhancement of the SSE voltage at room temperature is in a stark contrast to previous observations of the magnon–polaron SSE in YIG and other ferrite systems[7,9–11,13]. In fact, the observed enhancement in BiGa:LuIG is ~700% greater than that observed in a YIG film at room temperature (Fig. 2e, see the Methods section for the estimation of the magnon–polaron SSE enhancement). Moreover, the features of the resonant SSE enhancement are already present at very low-field values, as visible in the inset of Fig. 2b, where we can see a larger background SSE signal for $H < H_{TA}$.

Let us consider possible reasons for the observation of clear magnon–polaron SSE peaks at room temperature. As a result of the coupling between phonon (heat) and spin system, at the crossing of the magnon and phonon dispersions hybridized particles are formed, which are neither pure magnon or phonon but a magnon–polaron[20,21]. The lifetime of these magnon–polaron quasiparticles is associated to the ratio between the phonon and magnon lifetimes. In the theory of the magnon–polaron SSE[7,8], the peaks in the voltage are understood in terms of increased effects of the magnon–phonon coupling at

the magnetic-field values where the magnon and phonon dispersions just tangentially touch each other, resulting in the magnon–phonon hybridization over a larger volume in momentum space. This results in a resonant enhancement of the SSE when the acoustic quality of the crystal is larger than its magnetic quality, with the enhancement proportional to the ratio $\eta \propto |\tau_{ph}/\tau_{mag}| > 1$ between the phonon $\tau_{ph}$ and magnon $\tau_{mag}$ lifetimes[7,8]. In the case of a crystal having a better magnetic than acoustic quality ($\eta < 1$), dip structures instead of peaks in the SSE voltage are expected[8]. Therefore, according to the picture above, there are two possible scenarios to explain the larger magnon–polaron SSE peaks observed at room temperature: (1) an increased overlap over $k$-space at the touching fields ($H_{TA,LA}$) or (2) a larger ratio between the magnetic and acoustic quality of the crystal ($\eta \gg 1$). The first scenario can be discarded in our system, since the effect of larger magnetic compensation increases the curvature of the magnon dispersion (obtained in the next section), which results in a rather reduced overlap between the magnon–phonon dispersions at the touching points (see Supplementary Note 6). Then we must look at the scenario (2): it has been shown that the Bi substitution results in a decrease of

the magnon lifetime in YIG[22]. As a consequence, the larger ratio between the impurity scattering potentials, or $\eta$, can be expected, and therefore a greater enhancement of the lifetime of magnons by hybridization with the phonons at the touching points, resulting in a greater SSE enhancement. This interpretation is further supported by time-resolved Brillouin light-scattering (BLS) measurements (see Supplementary Note 3 for further details), showing that the magnon lifetime in BiGa:LuIG ($\tau_{mag} \sim$ 13 ns) is strongly decreased with respect to that of YIG ($\tau_{mag} \sim$ 50–75 ns)[23,24]. Further knowledge of the phonon lifetimes, $\tau_{ph}$, would be required in order to quantitatively discuss the magnitude of the resonant enhancement of the SSE.

We will now center the rest of our discussion mainly on the magnetic-field dependence of the magnon–polaron peaks and its relation to the dispersion characteristics. Let us now try to understand the magnitude of the magnetic fields required for the observation of the SSE peaks by considering the magnon and phonon dispersions. First, we consider linear TA and LA phonon dispersions $\omega_p = c_{TA,LA}k$, where the phonon velocities of the sample need to be considered. We have experimentally determined the phonon velocities using the optical spectroscopy method introduced by Hashimoto et al.[18], obtaining: $c_{TA}$ = $2.9 \times 10^3$ ms$^{-1}$ and $c_{LA}$ = $6.2 \times 10^3$ ms$^{-1}$ for TA and LA modes, respectively (these are fully consistent with those reported for a LuIG system with similar Bi, Ga doping)[18,25]. If we now assume the conventional magnon dispersion for a simple ferromagnet: $\omega_m = \gamma\mu_0 H + D_{ex}k^2$, as previously used for YIG[7], we can never explain the observed results without assuming an exceedingly large spin-wave stiffness parameter, with a value about four times higher than previously reported for YIG ($D_{ex} = 7.7 \times 10^{-6}$ m$^2$ s$^{-1}$)[7,26,27] and LuIG[28]. Even if we take into account the effect of Bi-doping, this can only explain a 1.4 times increase of the spin-wave stiffness magnitude, as shown in previous reports for similar Bi-doping in YIG[22]. Moreover, if we consider the expression of the spin-wave stiffness for a ferromagnet $D_{ex} \propto JSa^2$ (here $J$ corresponds to the exchange constant, $S$ the spin of the magnetic atoms, and $a$ the distance between neighboring spins), this estimation would imply an increased magnitude of $J$ despite the presence of nonmagnetic ion substitutions, contrary to expectations. This shows that the magnon dispersion of simple ferromagnets cannot capture the microscopic features of our system, therefore in the following, we will focus our attention on the ferrimagnetic ordering: effects of the Ga doping on the magnetic compensation, and its impact on the magnon dispersion characteristics of our system.

**Determination of the magnon dispersion**. Although the magnon dispersion characteristics of the YIG ferrimagnet have been previously studied[29–31], we are interested here in the effect of the degree of magnetic compensation on the magnon dispersion. As previously explained, the ferrimagnetic order in iron garnet systems typically arises from the different $Fe^{3+}$ occupation between the two magnetic sublattices, with the 3 to 2 ratio of tetrahedral (d) to octahedral (a) sites, as shown in Fig. 1d. Therefore, the degree of magnetic compensation can be modified by substitution of the magnetic $Fe^{3+}$ ions in the tetrahedral sites by nonmagnetic ones (i.e., Ga). To describe this effect, we consider the conventional Heisenberg Hamiltonian for a ferrimagnetic system[15,32], and express it as a function of the occupation numbers in the d and a sites. The expression of the Hamiltonian with the exchange and the Zeeman interaction terms is given below, where we have neglected dipolar and magnetic anisotropy interactions for

simplicity:

$$H = \frac{J_{ad}}{\hbar^2}\sum_{i,\delta}^{N_d} \mathbf{S}_{d,i}\mathbf{S}_{a,i+\delta} - \gamma\mu_0 H \sum_i^{N_d} S_{d,i}^z$$
$$+ \frac{J_{ad}}{\hbar^2}\sum_{j,\delta}^{N_a} \mathbf{S}_{d,j+\delta}\mathbf{S}_{a,j} - \gamma\mu_0 H \sum_j^{N_a} S_{a,j}^z, \tag{1}$$

here, the upper (lower) part of the Hamiltonian accounts for the tetrahedral (octahedral) sites, with $N_d$ ($N_a$) magnetic ions per unit volume. $J_{ad}$ is the nearest-neighbor inter-sublattice exchange (positive in the above expression), $\delta$ represents a vector connecting the nearest-neighbor a–d sites, $\mathbf{S}_x$ (x = a, d) are the spin operators (for a, d sites), $\mu_0 H$ is the external magnetic field, and $\gamma = g\mu_B/\hbar$ is the gyromagnetic ratio, with $g$ the spectroscopic splitting factor, $\mu_B$ the Bohr magneton, $\hbar$ the reduced Planck constant. Then, using the standard Holstein–Primakoff approximation[32,33], we can obtain the magnon dispersion relation as a function of the ratio of $Fe^{3+}$ ions occupying a and d sites (see Supplementary Note 4 for details of the derivation):

$$\omega_m = \gamma\mu_0 H + \frac{J_{ad}S(z_{da}\mu + z_{ad}\lambda)}{2\hbar}$$
$$\left\{ -\left(\frac{\mu - \lambda}{\lambda\mu}\right) \pm \left[\left(\frac{\mu - \lambda}{\lambda\mu}\right)^2 + \frac{4k^2a^2}{3\lambda\mu}\right]^{1/2} \right\}, \tag{2}$$

where $S = 5/2$ for $Fe^{3+}$, $a$ is the nearest-neighbor a–d distance, $z_{ad}$ ($z_{da}$) and $\lambda = N_a/N$ ($\mu = N_d/N$) correspond to the number of nearest-neighbors and occupation ratio of magnetic ions in octahedral (tetrahedral) sites, respectively, where $N$ is the total number of magnetic $Fe^{3+}$ ions per unit volume. Using the above expression, we calculate the magnon dispersion in the two sublattice ferrimagnet: $Lu_2Bi[D_xFe_{2-x}](D_yFe_{3-y})O_{12}$, where $D_x$ ($D_y$) denotes the nonmagnetic ions in octahedral (tetrahedral) sites with concentrations $x$ ($y$), the occupation ratio of the a (d) sites can be expressed as a function of $x$ ($y$) as $\lambda = 0.4\left(\frac{2-x}{2}\right)$ ($\mu = 0.6\left(\frac{3-y}{3}\right)$)[34]. Equation (2) can explain the magnetic-field values of previously observed magnon–polaron SSE in YIG[7] with $x = y = 0$ (no substitutions). Comparing our model with magnon–polaron SSE measurements in YIG at room temperature[10], we estimated the value of the exchange constant at 300 K, obtaining $J_{ad} = (4.3 \pm 0.2) \times 10^{-22}$ J (see Supplementary Note 5), which shows reasonable agreement with recently reported values by neutron scattering measurements ($J_{ad} = 4.65 \times 10^{-22}$ J)[35].

We now evaluate the effect of the nonmagnetic ion substitutions on the magnon spectrum, we can see that as the magnetic compensation of the system increases (larger $y$) the dispersion becomes gradually steeper (see Fig. 3a), with higher magnon frequencies for the same wave-vector values. When the system is fully compensated ($x = 0$, $y = 1$) a linear dispersion is obtained, as expected for an antiferromagnetic system.

To further test the validity of our model, we also performed wave-vector-resolved BLS spectroscopy measurements[19,36,37] to obtain the spin-wave dispersion of our system and compare it to our model. As shown in Fig. 3b, the peak frequency in BLS spectra for different wavenumbers measured at $\mu_0 H = 0.18$ T can be explained with the calculated magnon dispersion using $x = 0.101$ and $y = 0.909$. This composition is in close agreement with the one estimated by X-ray spectroscopy (see the Methods section). This result further proves the good agreement between the experiment and the theoretical model. Note that the small deviation of BLS plots at small $k$ region is due to dipole–dipole interaction, which is not considered in our model.

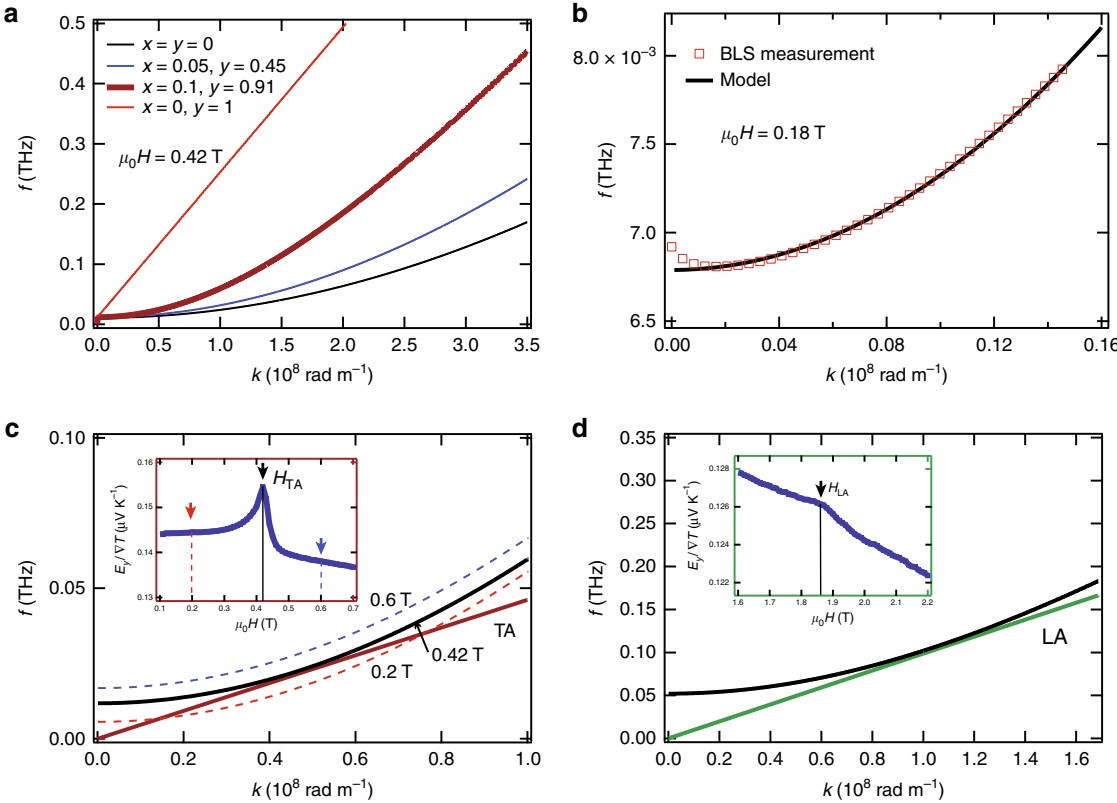

**Fig. 3** Effect of magnetic compensation on the spin-wave dispersion. **a** The results of the magnon dispersion ($2\pi f = \omega$) calculated using our model (Eq. (2)) for different concentrations of nonmagnetic ion substitutions ($x, y$), illustrating the effect of magnetic compensation in a ferrimagnetic system. A 90% preferential tetrahedral site occupation by Ga ions is assumed according to previous reports[17,50] and $J_{ad} = 4.3 \times 10^{-22}$ J, $a = 0.346$ nm (see Supplementary Note 5). **b** Comparison between the magnon dispersion measured experimentally by Brillouin light scattering and the theoretical dispersion for a ferrimagnet with nonmagnetic ion substitution of $x = 0.101$ (octahedral site occupation) and $y = 0.909$ (tetrahedral site occupation) (theoretical dispersion is vertically shifted to account for demagnetizing and anisotropy field contributions). **c, d** Magnon and phonon dispersions for $x = 0.101, y = 0.909, J_{ad} = 4 \times 10^{-22}$ J and $a = 0.346$ nm at magnetic fields of $\mu_0 H = 0.42$ T and $\mu_0 H = 1.86$ T depicting the condition for the observation of magnon–polaron spin Seebeck effect (SSE) enhancement by hybridization of magnon with transversal acoustic (TA) **c** and longitudinal acoustic (LA) **d** phonons in our system (insets show detail of the measured SSE voltage enhancement at the peak positions). In **c**, the magnon dispersion for three different field values, corresponding to the arrows in the inset, is shown

Now we are in a position to look into the magnetic-field dependence of the SSE voltage and the conditions for the observation of magnon–polaron in our system. The low magnetic-field value for the observation of the magnon–polaron SSE is attributed to an increased magnetic compensation which makes the magnon dispersion steeper compared with the non-doped case. This results in the touching condition for the magnon and phonon dispersions at lower magnetic fields than that of YIG. Figure 3c, d shows the comparison of the magnon and phonon dispersions with the above estimated composition at the magnetic fields $H_{TA}$ and $H_{LA}$ for which the SSE peaks are observed. In Fig. 3c, we can distinguish three different regions of the SSE response with respect to the magnitude of the magnetic field: for $H < H_{TA}$, the SSE has both magnonic and phononic contributions (possibly from the crossing points of the dispersions), at $H = H_{TA}$ the SSE is resonantly enhanced showing peak structures, due to the increased effect of magnon–phonon hybridization at the touching condition between the dispersions and at $H > H_{TA}$, a shift of the SSE voltage background signal can be observed, which is likely due to the fact that the magnon and TA–phonon dispersions do not cross at any point of the $\omega - k$ space and the contribution from the TA phonons is suppressed. The presence of this shift in the SSE voltage indicates that the contribution from magnon–phonon coupling effects can be already sensed at

magnetic fields values much lower than that of the touching condition (see inset of Fig. 2b).

**Temperature dependence of magnon–polaron SSE.** Let us now investigate the temperature dependence of the magnon–polaron SSE. Magnetization measurements as a function of temperature show that the increased magnetic compensation, due to the presence of Ga substitution, results in the reduction of the saturation magnetization and ferrimagnetic ordering temperature ($T_c$) in LuIG with $T_c = 401.6 \pm 0.2$ K (see Fig. 4b). Figure 4a shows the result of the magnetic-field-dependent SSE voltages at different temperatures, we can see that the magnitude of the SSE decreases close to the transition temperature as previously reported on YIG[38], and eventually, at $T \approx 400$ K, the magnon-driven SSE is suppressed, showing only a weak voltage with a paramagnetic-like magnetic-field dependence. The magnon–polaron SSE is gradually weakened as the temperature increases toward the transition temperature. The peaks are strongly suppressed at temperatures well below the transition temperature, with the maximum temperature for observation of the peaks: $T = 380$ K for TA and $T = 315$ K for LA phonons, possibly due to the much weaker amplitude of the LA peaks compared with the TA peaks (see Fig. 2c, d) near the transition temperature. Here, we will now focus on the temperature

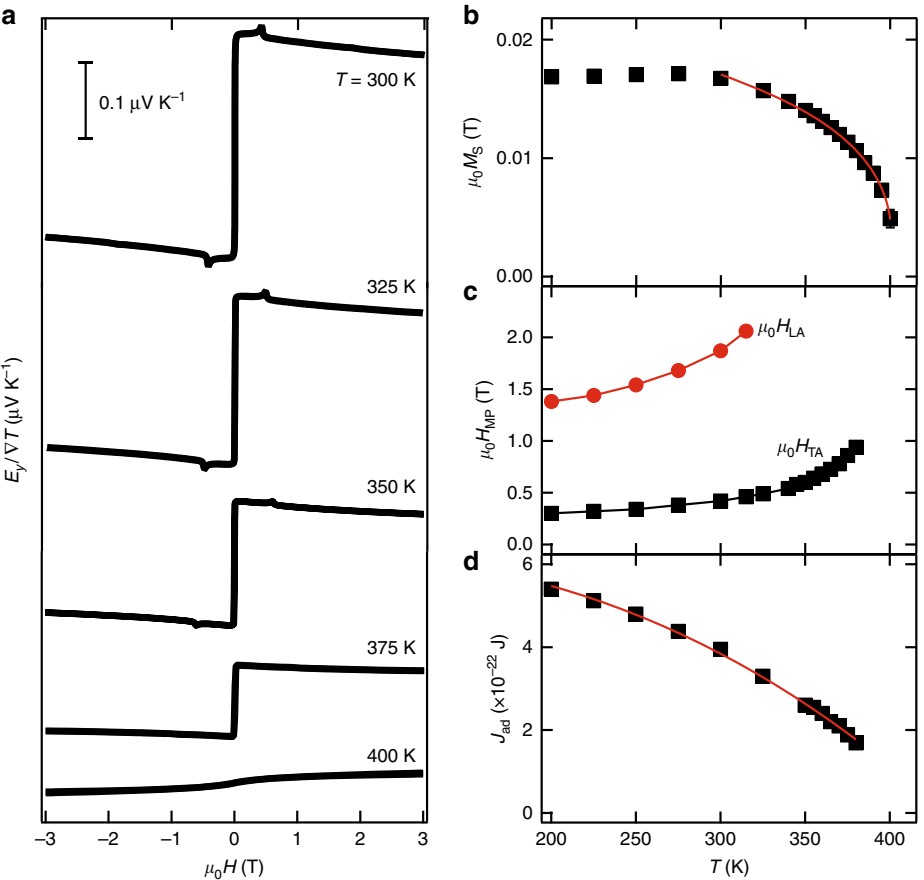

**Fig. 4** Temperature-dependent magnetization and heat-driven spin transport properties of BiGa:LuIG film. **a** Magnetic-field dependence of the measured spin Seebeck effect (SSE) at different temperatures (the data are vertically offset for clarity). Scale bar, $10^{-7}$ VK$^{-1}$. **b** Saturation magnetization ($\mu_0 M_S$) as a function of temperature. Red line shows fitting to $\mu_0 M_S \propto (T_c - T)^\beta$, with $T_c = 401.6 \pm 0.2$ K and $\beta = 0.305 \pm 0.007$. **c** Temperature dependence of the magnetic-field magnitude for the observation of the magnon–polaron SSE, $H_{TA}$ (black squares), and $H_{LA}$ (red circles) (lines interconnecting experimental points are for visual guide). **d** Temperature dependence of the intersite exchange integral ($J_{ad}$) estimated from the magnon–polaron fields in **c** and the the condition for tangential touching of the magnon and phonon dispersions at different temperatures. Red line shows fitting to $J_{ad} = J_0(1 - \eta T^{5/2})$, with $J_0 = (6.41 \pm 0.05) \times 10^{-22}$ J

dependence of the magnetic field at which the magnon–polaron peaks are observed, which is shown in Fig. 4c: we can see that the magnitude of the magnetic fields required for the magnon–polaron to appear increase with the temperature. This trend cannot be understood by the elastic properties of the garnet system. Previous reports show a nearly constant phonon velocity upon increasing temperature above room temperature[39], with just a slight decrease of <5% per 100 K increase, which would imply a slight decrease of the magnon–polaron magnetic fields upon increasing temperature[7], opposite to our observation. In contrast, the observed temperature dependence of the magnon–polaron magnetic fields can be understood by the softening of the magnon dispersion upon increasing temperature; in the previously obtained magnon dispersion (Eq. (2)), most of the parameters are temperature independent, except for the intersite exchange energy $J_{ad}$. The observed temperature dependence of the magnon–polaron magnetic field can be explained in terms of the magnitude of $J_{ad}$, which decreases around the transition temperature, in agreement with the previous observations in YIG and other ferrimagnets[30,40,41].

The temperature dependence of $J_{ad}$ estimated from the touching condition between magnon and phonon dispersions is shown in Fig. 4d, the obtained dependence can be understood in terms of a temperature-dependent exchange energy, with an expression similar to that used in Ref. [41]: $J_{ad} = J_0(1 - \eta T^{5/2})$,

with $J_0 = (6.41 \pm 0.05) \times 10^{-22}$ J, which is obtained from considering the effect of magnon–magnon interactions in a ferromagnet[42]. These results possibly suggest that the magnon–magnon interactions might play a role in the magnon–polaron SSE in the temperature region studied here.

## Discussion

In this study, we reported the resonant enhancement of SSE in a partially compensated ferrimagnet. Sharp peaks were observed in SSE voltage at room temperature and low magnetic fields. The resonant enhancement of SSE is 700% greater than that observed in YIG films, atributable to reduced magnon lifetime of BiGa: LuIG in comparison with YIG, which results in larger reinforcement of the magnon lifetimes affected by the phonon system via the hybridization.

The observed resonant enhancement of SSE at low magnetic fields is attributable to steeper magnon dispersion caused by the increased magnetic compensation of the system. Our results show the possibility to tune the spin-wave dispersion by chemical doping, which allows exploring magnon–phonon coupling effects at different regions of the spin-wave spectrum. The value of the intersite exchange parameter ($J_{ad}$) was also estimated with values in reasonable agreement with previous studies. This fact shows the potential of the SSE as a table-top tool to investigate the spin-wave dispersion characteristics in comparison with other more

expensive and less accessible techniques, such as inelastic neutron scattering[9,31].

## Methods

**Sample characterization**. We used epitaxial $Lu_2BiFe_4GaO_{12}$ (4 μm) and YIG (2.5 μm) films grown by liquid phase epitaxy on $Gd_3Ga_5O_{12}$ [001] and [111]-oriented substrates, respectively. The samples have been characterized by X-ray diffraction (XRD), transmission electron microscopy (TEM), and scanning TEM (STEM), see Supplementary Note 1 for more details. XRD was performed in a Bruker D8 Discover high-resolution diffractometer, TEM was performed in a EM-002B microscope from Topcon ($LaB_6$ cathode electron gun operated at 200 kV), and STEM was performed in a JEM-ARM200F microscope from JEOL (cold cathode field emission (FE) electron gun operated at 200 kV). The composition of the $Lu_2BiFe_4GaO_{12}$ (BiGa:LuIG) film was determined using wavelength dispersive X-ray spectroscopy (WDX). Magnetization measurements show a saturation magnetization $\mu_0 M_S = 0.017$ T at 300 K and a transition temperature of $T_c \approx 402$ K. A 5-nm-Pt layer was deposited at room temperature by (DC) magnetron sputtering in a sputtering system QAM4 from ULVAC, with a base pressure of $10^{-5}$ Pa. Before forming the Pt films, the surface of the BiGa:LuIG was mechanically polished with alumina powder with a diameter of 0.05 μm. The resultant BiGa: LuIG/Pt bilayers have clean and atomically sharp interfaces as confirmed by TEM measurements (see Supplementary Note 1 for more details).

**Spin Seebeck effect**. The SSE measurements were performed in a physical property measurement (PPMS) Dynacool system of Quantum Design, Inc., equipped with a superconducting magnet with fields of up to 9 Tesla. The system allows for temperature-dependent measurements from 2 to 400 K. For the SSE measurements, the sample is placed between two plates made of AlN (good thermal conductor and electrical insulator): a resistive heater is attached to the upper plate and the lower plate is in direct contact with the thermal link of the cryostat, providing the heat sink. The temperature gradient is generated by applying an electric current to the heater, while the temperature difference between the upper and lower plate is monitored by two E-type thermocouples connected differentially. The samples are contacted by Au wire of 25-μm diammeter. To minimize thermal losses, the wires are thermally anchored to the sample holder. The thermoelectric voltage is monitored with a Keithley 2182 A nanovoltmeter. The sample dimensions for the SSE measurements were $L_y = 6$ mm, $L_x = 2$ mm, and $L_z = 0.5$ mm.

**Estimation of the magnon–polaron peak enhancement**. The magnitude of the SSE enhancement at the magnon–polaron peaks was estimated from the ratio $MS_{MP}(\%) = \frac{\delta S_{MP}}{S_0} \times 100$, where $S_0$ is the extrapolated background SSE coefficient at the peak position and $\delta S_{MP} = S(H_{MP}) - S_0$ is the magnon–polaron (MP) peak height (see Fig. 2e), with $S(H_{MP})$ the SSE coefficient at the MP (TA or LA) peak. The spin Seebeck coefficient was previously defined as $S = \frac{E_y}{\nabla T}$ (this is equivalent to expressing it in terms of the heat flow $j_q$, when the thermal conductivity of the sample $\kappa$ is known $\left(\frac{E_y}{\nabla T} \equiv \kappa \frac{E_y}{j_q}\right)$[43].

By using the ratio $MS_{MP}$ instead of the absolute coefficient $S$, we can remove the dependence of the estimated MP enhancement on possible extrinsic effects due to differences in materials thermal properties or experimental conditions (i.e., thermal contact differences) when comparing the MP enhancement between different samples. The corresponding estimated enhancements are $MS_{TA}$(BiGa:LuIG) = 10.21% and $MS_{TA}$(YIG) = 1.27%. Then, the enhancement in BiGa:LuIG relative to that of YIG is estimated as: $\frac{MS_{TA}(\text{BiGa:LuIG}) - MS_{TA}(\text{YIG})}{MS_{TA}(\text{YIG})} \times 100 = 700\%$.

**Magnetization**. The magnetization measurements were performed using the vibrating sample magnetometer (VSM) option of a PPMS system by Quantum Design, Inc. The temperature-dependent magnetization measurements were obtained by performing isothermal M–H loops at each temperature and extracting the saturation magnetization from the value measured at 30 mT for each temperature. A comparison of the saturation magnetization ($M_S$) with that of the YIG film[44,45] clearly shows that the magnetization saturation, $M_S$ and the magnetic ordering temperature, $T_C$ are reduced as a result of the increased magnetic compensation. We have also confirmed the absence of low magnetic-field effects in the SSE induced by surface magnetic anisotropies[46–49] (see Supplementary Note 2 for more details about the magnetic characterization).

**Magnon dispersion**. The Brillouin light-scattering (BLS) measurements were performed using an angle-resolved Brillouin light-scattering setup. Brillouin light scattering is an inelastic scattering of light due to magnons. As a result, some portion of the scattered light shifts in the frequency equivalent to that of magnon. The scattered light is introduced to multi-pass tandem Fabry-Perot interferometer to determine the frequency shift. Wavenumber resolution was realized by collecting the back-scattered light from the sample by changing the incident angle ($\theta_{in}$). As a result of conservation of wavenumber of magnon ($k_m$) and light ($k_l$), the wavenumber of magnon is determined as $k_m = 2k_l \sin(\theta_{in})$. All the spectrum was obtained at room temperature.

## Data availability

The data that support the findings of this study are available from the corresponding author upon reasonable request.

## Code availability

The code used to calculate the magnon dispersions in Fig. 3 is available from the corresponding author upon reasonable request.

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

## Acknowledgements

The authors thank S. Ito from the Institute for Materials Research, Tohoku University, for performing TEM and STEM on our samples. We thank J. Barker for fruitful discussions. This work was supported by ERATO "Spin Quantum Rectification Project" (Grant no. JPMJER1402) and Grant-in-Aid for Scientific Research on Innovative Area, "Nano Spin Conversion Science" (Grant no. JP26103005), Grant-in-Aid for Scientific Research (S) (Grant no. JP19H05600), Grant-in-Aid for Research Activity Start-up (No. JP19K21031) from JSPS KAKENHI, JSPS Core-to-Core program "the International Research Center for New-Concept Spintronics Devices" Japan, and Deutsche Forschungsgemeinschaft (DFG) project HI 380/29-1 "Magnon currents for future spintronic applications" within the JSPS Core-to-Core Joint International Research and Collaborative Center and the NEC Corporation.

## Author contributions

R.R. performed the measurement, analyzed the data, and developed the theoretical model with input from T.H., Y.H., T.K. and E.S. P.F., A.J.E.K., T.H., V.I.V. and A.A.S. performed the BLS measurements. T.K. performed the SSE measurements in YIG. R.R., Y.H. and E.S. planned and supervised the study. R.R. wrote the paper with review and input from T.H., T.K., B.H. and E.S. All authors discussed the results and commented on the paper.

## Competing interests

The authors declare no competing interests.
