## [Peer Review File · Nature Communications]

Reviewers' Comments:

Reviewer #1:

Remarks to the Author:

The authors reported on the room temperature low-field resonant enhancement of SSE in YIG and doped systems. A large enhancement of 700% has been achieved in the doped sample Lu₂BiFe₄GaO₁₂. The results obtained are sound and valuable for consideration in this journal. However, several uncertain issues need to be addressed properly before the paper can be accepted.

1. No structural information/characterization/data like XRD, SEM, TEM were presented for the films studied in this work? Since the SSE is related to the surface and interface properties, it is essential to report these data.

2. No details of the magnetic characterization of Lu₂BiFe₄GaO₁₂ were reported in this paper. The authors stated $M_s = 0.024$ T at 300 K and $T_c \sim 402$ K, but no magnetic data was shown to support their claim. I would suggest adding the magnetic data (M-H and M-T data) of this sample to Supporting information in which the magnetic data of the YIG film was already presented. It is essential to understand how the doping altered the magnetic behavior of Lu₂BiFe₄GaO₁₂ relative to YIG.

It has been reported that YIG has a complex magnetic structure, with the out-of-plane spin configuration in the surface layer and the in-plane spin alignment in the interior layer (Phys. Rev. B 92, 014415 (2015); J. Appl. Phys. 116, 153705 (2014); Phys. Rev. B 94, 024405 (2016); Scientific Reports 7, 13316 (2017)). This property could depend on the thickness of the film, as well as chemical doping. Several studies have shown that bulk and surface perpendicular magnetic anisotropies play important roles in the SSE in YIG/Pt systems (Phys. Rev. B 92, 014415 (2015); Scientific Reports 7, 13316 (2017)). Unfortunately, such effects were not studied or discussed in this manuscript.

It is essential to know how the doping (Lu,Bi,Ga) altered the magnetic anisotropy and its association with the SSE?

Also, effects of Pt coating on the magnetic properties of the YIG and doped-YIG films might be different which should also be studied and discussed in this work.

3. Surface roughness of the magnetic layer could affect SSE signal considerably. So, it is unclear how the doping affected the sample's surface topology and hence the observed SSE. Some details of discussion should be given to address this point.

4. Effects of magnetic damping and spin diffusion length on the SSE could also be considerable in both systems. Experiments/comments on how the doping altered these parameters and consequently the SSE would be welcome.

5. Data in figure 4 could have been explained more clearly. It is clear to see that H_{la} increased until $T \sim 315$ K at which the magnetization started to drop on increasing the temperature, while H_{ta} kept increasing until the phase transition finished. More discussions should be given to clarify this. Perhaps it would be useful to substrate these two fields and relate the resulting field to relative contributions of LA and TA components to the observed effect near the magnetic phase transition.

6. While SSE measurements were performed at $T > 200$ K (Fig. 4), it has been reported that there is a spin reorientation below 175K due to the single ion anisotropy of Fe⁺² ions (ACTA Physica Sinica 38, 11 (1989)). This has been shown to affect the magnetic anisotropy and hence the SSE in YIG/Pt (Scientific Reports 7, 13316 (2017)). Testing if this would also occur in the present films would also be interesting.

7. There are some typos in the manuscript, which should be checked and corrected.

Reviewer #2:

Remarks to the Author:

The authors reported an enhancement of spin Seebeck effect (SSE) at particular magnetic fields when the magnon and phonon dispersions touch in a partially compensated garnet, $\text{Lu}_2\text{BiFe}_4\text{GaO}_{12}$. The same research group did the pioneering work on the exactly same phenomenon in $\text{Y}_3\text{Fe}_5\text{O}_{12}$ or YIG and came up with the explanation based on the magnon-phonon hybridization mechanism. In this work, by introducing Ga which substitutes the tetrahedral sites of Fe^{3+} , the number of antiferromagnetically aligned Fe^{3+} ions on both tetrahedral and octahedral sites are nearly equal. They observed a couple consequences: 1. a factor of 7 increase in the low-field SSE peak compared to that in YIG; 2. the resonant field shift towards to a lower value. The authors proposed a model to explain the resonant field shift by the change of the magnon dispersion in compensated garnets. The experimental results are sound and the magnon-phonon hybridization itself is a very interesting problem in physics and materials science. However, I do not recommend accepting for the following reasons.

1. This work does not represent any conceptual breakthrough as they first reported the phenomenon in PRL. The same basic anomalies in SSE at two particular magnetic fields but lower than those in YIG. They proposed that the quadratic to linear dispersion crossover as the garnet is more compensated is responsible for the lower resonant fields. This is certainly one possibility. However, the authors completely neglect possible changes in LA and TA phonon dispersion relations when Ga atoms are introduced. In principle, if such substitution leads to lower sound velocities, it can also cause the downshift in the resonant fields. This possibility was not discussion.
2. The authors discussed two possibilities responsible for the SSE voltage enhancement: larger overlap between two dispersions; longer phonon lifetime relative to magnon lifetime. The authors argued against the first one but did not present any experimental evidence to show the phonon/magnon lifetime ratio is larger in the compensated garnet.
3. Related to point 2, in the Suppl. Mat., the authors attempted to show that a more linear magnon dispersion produces a smaller overlap with the phonon dispersion. This is somewhat counter-intuitive. In the limiting case, the overlap between two lines is infinitely long.
4. BLS results were presented for a small range of wavevectors, and parameters such as exchange constant were obtained. However, more details would be helpful. For example, both phonons and magnons can produce Brillouin peaks in the same frequency and wavevector range. Did they observe phonon peaks? If not, why phonon peaks are absent? If yes, it would be good to present both.
5. The temperature dependence of the resonant fields was attributed to the change in the exchange constant. How about possible change in phonon dispersion over the same temperature range?
6. There is no ref. 38 as they quoted in the manuscript.

Reviewer #3:

Remarks to the Author:

This work reports on the resonant enhancement of the spin-Seebeck effect (SSE) due to magnon-phonon coupling at a certain value of magnetic field. The enhancement occurs when a specific mode of the magnon dispersion, which is more or less parabolic, shifts upwards in energy with an external applied magnetic field, and becomes tangential with an acoustic phonon mode, which is linear and field-independent. The hybridization between phonon and magnon modes then covers a wider range of momentum space. The theory for the effect, worked out in Ref. 8, is based on the need for heat to couple from the phonon system to the magnon system. This, according to the

authors, is proportional to a factor η defined as the ratio of the phonon lifetime to the magnon lifetime. The paper notices that the resonant enhancement of the SSE is much larger in a Lu-Bi-Ga-Fe garnet (LBGFG) than in the more well-known yttrium-iron garnet (YIG). This, in turn, they attribute to the lower magnon lifetime in LBGFG, and I agree. The paper further points out that the dispersions of the magnon branches in YIG deviate strongly from parabolic.

Basically, because I think that this work opens the prospect of using SSE in the analysis of hybridized modes, my opinion is that the results are new enough and important enough to be published in Nat. Comm. However, I would like to point a few things out.

1. The magnitude difference between the result on YIG and on LBGFG could have a contribution that is not intrinsic to the magnon-phonon coupling but simply to the way the data are plotted. The reason for this argument is the difference in thermal conductivity between YIG and LBGFG, which in turn is due to alloy scattering of phonons. Both films are grown on GGG. The heat flux creates a temperature drop across both the GGG and the YIG or LBGFG films. The data are presented in $\mu\text{V}/\text{K}$, where K is the unit for the temperature difference across the structure. But since YIG is more conductive than LBGFG, a larger fraction of the temperature drop across the structure occurs across the film in the LBGFG sample than in the YIG sample. The thermal driving force is therefore higher in that sample, irrespectively of the resonance. Is the effect still present when the SSE effect is plotted as microvolts per unit of heat flux for both samples? This is a quick check the authors should have all data to carry out.

2. The factor η in the theory raises questions. I suppose the important parameter is the transfer of energy and momentum between magnons and phonons. Why consider the total lifetime of these quasi-particles? Isn't what matters the fraction of collisions of magnons with phonons to those of magnons with everything else? Secondly, the two terms should be accessible experimentally for YIG from existing neutron data: these lifetimes are given by the bandwidth in energy of the measured magnon and phonon dispersions. Thirdly, is a resonant condition occurs, that too should be visible experimentally via neutron scattering, similarly to Delaire's work on phonons (Nature Materials volume 10, pages 614–619 (2011)).

3. The manuscript spends a lot of effort on calculating the magnon dispersions, which deviate considerably from parabolic. While important to the discussion here, this was known for a long time (Ref. 33 and Plant, J. Phys. C. 10, 4805-4814 (1977)).

4. type p4 line 3, "... , there ARE two possible..."

Reviewers' comments:

Reviewer #1 (Remarks to the Author):

The authors reported on the room temperature low-field resonant enhancement of SSE in YIG and doped systems. A large enhancement of 700% has been achieved in the doped sample Lu₂BiFe₄GaO₁₂. The results obtained are sound and valuable for consideration in this journal. However, several uncertain issues need to be addressed properly before the paper can be accepted.

We thank the reviewer for the positive remarks and the comments. Below we provide a point-by-point response to all raised criticisms.

1. No structural information/characterization/data like XRD, SEM, TEM were presented for the films studied in this work? Since the SSE is related to the surface and interface properties, it is essential to report these data.

We have measured X-ray diffraction (XRD) and transmission electron microscopy (TEM) for the GGG/Lu₂BiFe₄GaO₁₂/Pt bilayer studied in the present manuscript. The results show that the Lu₂BiFe₄GaO₁₂ (BiGa:LuIG) film grows epitaxially on the GGG(001) substrate with perfect lattice matching and atomically sharp interface between film and substrate (see Supplementary Note 1 and Supplementary Figure 1 for details).

We have also added a comment in the “Sample characterization” section in Methods indicating that XRD and TEM measurements were performed and that the details are available in the supplementary information (Supplementary Note 1).

2. No details of the magnetic characterization of Lu₂BiFe₄GaO₁₂ were reported in this paper. The authors stated $M_s = 0.024$ T at 300 K and $T_c \sim 402$ K, but no magnetic data was shown to support their claim. I would suggest adding the magnetic data (M-H and M-T data) of this sample to Supporting information in which the magnetic data of the YIG film was already presented. It is essential to understand how the doping altered the magnetic behavior of Lu₂BiFe₄GaO₁₂ relative to YIG.

Following the comment, we have now included in the Supplementary Information more details about the magnetic characterization of the sample (Supplementary Note 2), where representative results of the magnetic field dependence of the magnetization measured at several temperatures are shown (we have also added a note correspondingly in the manuscript in page 3, line 64).

These data was previously used to obtain the temperature dependence of the saturation magnetization M_s , shown in Fig. 4b of the main text. We have now also included a comparison of the temperature dependence of M_s between BiGa:LuIG and YIG in the supplementary information (Supplementary Figure 2e), where it can be clearly seen that the doping causes a reduction of the magnetization and magnetic ordering temperature with values of $\mu_0 M_s(300\text{ K}) = 0.017$ T and $T_c = 402$ K, respectively. These are much lower than the respective values for YIG, with $\mu_0 M_s(300\text{ K}) = 0.178$ Tesla and $T_c = 553$ K as reported by K. Uchida et al. PRX **4** 041023 (2014), APL **106** 052405 (2015) and T. Kikkawa et al. PRB **92** 064413 (2015).

Note that the value of the saturation magnetization for the Lu₂BiFe₄GaO₁₂ film ($\mu_0 M_s(300\text{ K}) = 0.017$ T) has been corrected after considering the measured thickness of the film from the TEM measurements, we have modified figure 4b in the main text accordingly.

It has been reported that YIG has a complex magnetic structure, with the out-of-plane spin configuration in the surface layer and the in-plane spin alignment in the interior layer (Phys. Rev. B 92, 014415 (2015); J. Appl. Phys. 116, 153705 (2014); Phys. Rev. B 94, 024405 (2016); Scientific Reports 7, 13316 (2017)). This property could depend on the thickness of the film, as well as chemical doping. Several studies have shown that bulk and surface perpendicular magnetic anisotropies play important roles in the SSE in YIG/Pt systems (Phys. Rev. B 92, 014415 (2015); Scientific Reports 7, 13316 (2017)). Unfortunately, such effects were not studied or discussed in this manuscript.

It is essential to know how the doping (Lu,Bi,Ga) altered the magnetic anisotropy and its association with the SSE?

Also, effects of Pt coating on the magnetic properties of the YIG and doped-YIG films might be different which should also be studied and discussed in this work.

We agree with the reviewer that bulk and surface perpendicular anisotropies can play important roles in the SSE of YIG/Pt system at low magnetic fields, manifesting as differences in the magnetic field dependence between SSE voltage and magnetization loops in the low magnetic field region ($|H| < 100$ Oe). The effect is pronounced for a YIG slab, but it gradually decreases as the YIG film thickness is reduced [Phys. Rev. B 92, 014415 (2015)].

In order to check the absence of this effect in BiGa:LuIG(4 μm)/Pt, we have measured the magnetization and SSE at low magnetic fields. As shown in Figure 1 below, the magnetic field dependence of both magnetization and SSE closely follow each other, indicating that the above effects are negligibly small in our system. We have included a comment regarding this point in the ‘‘Magnetization’’ section in Methods and introduced the comparison between magnetization and SSE in the supplementary information (Supplementary Note 2).

Moreover, from the comparison between the hysteresis loops for the SSE and magnetization it is clear that the Pt coating plays no role in the magnetic properties of the garnet film. Since the magnetization and SSE measurements were performed in two different pieces of the same sample, one with no Pt coating (magnetization) and the other one with a 5nm Pt layer (SSE). Both samples have the same magnetic field dependence as shown below, therefore ruling out effects of Pt coating on the magnetic properties of the garnet.

Figure 1. Comparison between the magnetization and the SSE at 300 K for BiGa:LuIG (4 μm)

3. Surface roughness of the magnetic layer could affect SSE signal considerably. So, it is unclear how the doping affected the sample's surface topology and hence the observed SSE. Some details of discussion should be given to address this point.

We have performed X-ray reflectivity (XRR) and high-resolution transmission electron microscopy measurements, as mentioned in comment 1 above and shown in Supplementary Figure 1. The measured XRR shows that the BiGa:LuIG/Pt has a sharp interface with small roughness, moreover the high-resolution scanning TEM shows an atomically sharp interface (see Supplementary Figure 1i). Given the atomic sharpness of the BiGa:LuIG/Pt interface in our system we can rule out effects from surface roughness at the interface.

4. Effects of magnetic damping and spin diffusion length on the SSE could also be considerable in both systems. Experiments/comments on how the doping altered these parameters and consequently the SSE would be welcome.

We thank the reviewer for the insightful comment. We have investigated the magnon lifetime of the BiGa:LuIG garnet film by time-resolved Brillouin light scattering (BLS) measurements. We obtained a magnon lifetime for the $\text{Lu}_2\text{BiFe}_4\text{GaO}_{12}$ film of $\tau_m(\text{BiGa:LuIG}) \approx 13$ ns, showing a strong reduction in comparison to the one measured for a YIG film with $\tau_m(\text{YIG}) \approx 50$ to 75 ns [S. Rezende et al. Phys. Rev. B 89, 014416 (2014) and A. A. Serga et al. Nat. Comm. 5, 3452 (2014)].

This further supports our interpretation that the larger enhancement by the magnon-polaron SSE in BiGa:LuIG with respect to that of YIG is due to a decrease of the magnon lifetime.

We have added this information in the supplementary information (Supplementary Note 3), we have also added a comment in the manuscript indicating that the estimation of the magnon lifetime in BiGa:LuIG by time-resolved BLS further supports our interpretation (page 4, lines: 106-110).

5. Data in figure 4 could have been explained more clearly. It is clear to see that $H_{\parallel a}$ increased until $T \sim 315$ K at which the magnetization started to drop on increasing the temperature, while $H_{\perp a}$ kept increasing until the phase transition finished. More discussions should be given to clarify this. Perhaps it would be useful to substrate these two fields and relate the resulting field to relative contributions of LA and TA components to the observed effect near the magnetic phase transition.

The magnitude of the magnon-polaron (MP) anomaly for the longitudinal acoustic (LA) phonon is very weak in the studied region. This can be seen in Fig. 2d of the main text (also reproduced below). The MP-SSE for the LA phonon only accounts for a signal increase of about 0.6 nV/K with respect to the background SSE signal (~ 125.6 nV/K) at 300 K (see figure below), and given the fact that the magnitude of the MP anomalies tends to decrease as we approach the transition temperature, as seen in Figure 4a of the main text, we cannot electrically detect the MP-SSE for the LA phonon at temperatures $T > 315$ K.

Unfortunately, we cannot perform the proposed analysis near the transition temperature region since the LA peaks cannot be detected. We have added a comment correspondingly in page 8, lines 219-220 of the main text.

Figure 2. Detail of the magnon-polaron SSE (MP SSE) for the longitudinal acoustic (LA) phonons at 300 K.

6. While SSE measurements were performed at $T > 200$ K (Fig. 4), it has been reported that there is a spin reorientation below 175K due to the single ion anisotropy of Fe²⁺ ions (ACTA Physica Sinica 38, 11 (1989)). This has been shown to affect the magnetic anisotropy and hence the SSE in YIG/Pt (Scientific Reports 7, 13316 (2017)). Testing if this would also occur in the present films would also be interesting.

We have performed SSE measurements between 125 K and 200 K as shown in the figure below. We can see that there is no voltage drop around 175 K for our film, in contrast to that reported by V. Kalappatil et al. in Scientific Reports 7, 13316 (2017).

Figure 3. (a) Magnetic field dependence of the SSE at different temperatures in the region $125 < T$ (K) < 200 . (b) Magnitude of the SSE obtained from the intercept of the high field SSE voltage ($\mu_0H > 2$ T) extrapolated to zero field.

7. There are some typos in the manuscript, which should be checked and corrected.

We have carefully revised the manuscript and corrected the typos where found.

Reviewer #2 (Remarks to the Author):

The authors reported an enhancement of spin Seebeck effect (SSE) at particular magnetic fields when the magnon and phonon dispersions touch in a partially compensated garnet, Lu₂BiFe₄GaO₁₂. The same research group did the pioneering work on the exactly same phenomenon in Y₃Fe₅O₁₂ or YIG and came up with the explanation based on the magnon-phonon hybridization mechanism. In this work, by introducing Ga which substitutes the tetrahedral sites of Fe³⁺, the number of antiferromagnetically aligned Fe³⁺ ions on both tetrahedral and octahedral sites are nearly equal. They observed a couple consequences: 1. a factor of 7 increase in the low-field SSE peak compared to that in YIG; 2. the resonant field shift towards to a lower value. The authors proposed a model to explain the resonant field shift by the change of the magnon dispersion in compensated garnets. The experimental results are sound and the magnon-phonon hybridization itself is a very interesting problem in physics and materials science. However, I do not recommend accepting for the following reasons.

We thank the reviewer for acknowledging the quality of the results and highlighting the interest of the topic. We address below his/her criticisms and concerns point-by-point.

1. This work does not represent any conceptual breakthrough as they first reported the phenomenon in PRL. The same basic anomalies in SSE at two particular magnetic fields but lower than those in YIG. They proposed that the quadratic to linear dispersion crossover as the garnet is more compensated is responsible for the lower resonant fields. This is certainly one possibility. However, the authors completely neglect possible changes in LA and TA phonon dispersion relations when Ga atoms are introduced. In principle, if such substitution leads to lower sound velocities, it can also cause the downshift in the resonant fields. This possibility was not discussion.

We understand the reviewer concern that in order to support our interpretation (lower magnetic fields for the magnon-polaron SSE being due to the effect of increased magnetic compensation on the magnon dispersion) we also need to consider the effect of doping on the phonon dispersion. This has been experimentally addressed by measuring the phonon dispersion using the optical measurement method introduced in the previous paper by some of the co-authors [Y. Hashimoto *et al.* Nat. Commun. **8**, 15859 (2017)]. The obtained phonon velocities are fully consistent with those reported in the above paper for a LuIG film with similar Bi,Ga doping.

We apologize if this was not clear in the previous version of the manuscript, we have now added a comment stressing the fact that we experimentally determined the phonon velocities in page 4, lines 116-120 of the manuscript.

2. The authors discussed two possibilities responsible for the SSE voltage enhancement: larger overlap between two dispersions; longer phonon lifetime relative to magnon lifetime. The authors argued against the first one but did not present any experimental evidence to show the phonon/magnon lifetime ratio is larger in the compensated garnet.

We thank the reviewer for the insightful comment. We have investigated the magnon lifetime of the BiGa:LuIG garnet film by time-resolved Brillouin light scattering (BLS) measurements. We obtained a magnon lifetime for the Lu₂BiFe₄GaO₁₂ film of $\tau_m(\text{BiGa:LuIG}) \approx 13$ ns, showing a strong reduction in comparison to the one measured for a YIG film with $\tau_m(\text{YIG}) \approx 50$ to 75 ns [S. Rezende *et al.* Phys. Rev. B **89**, 014416 (2014) and A. A. Serga *et al.* Nat. Comm. **5**, 3452].

This further supports our interpretation that the larger enhancement by the magnon-polaron SSE in BiGa:LuIG, with respect to that of YIG, is due to a decrease of the magnon lifetime. We have added this information in the supplementary information (Supplementary Note 3), we have also added a comment in the manuscript indicating that the estimation of the magnon lifetime in BiGa:LuIG by time-resolved BLS further supports our interpretation (page 4, lines: 106-110).

3. Related to point 2, in the Suppl. Mat., the authors attempted to show that a more linear magnon dispersion produces a smaller overlap with the phonon dispersion. This is somewhat counter-intuitive. In the limiting case, the overlap between two lines is infinitely long.

We thank the reviewer for the comment and apologize for the confusion. In the Supplementary Note 6 we perform a power expansion of the calculated magnon dispersion relation at the touching condition to evaluate the degree of overlap.

At the touching condition ($k = k_{\text{touch}}$) both magnon and phonon velocities are equal ($\frac{\partial\omega_m}{\partial k} = \frac{\partial\omega_p}{\partial k} = c_{TA,LA}$), therefore the degree of overlap can be evaluated by the quadratic component (k^2) of the power expansion, which should be smaller to have a larger overlap between the dispersions.

We observed that the k^2 -component of the power expansion is actually increased with increasing level of Ga-substitution (magnetic compensation): $0 < y < 1$. This indicates that, counter-intuitively, the overlap over k -space for the Ga-doped sample is smaller than in the non-doped one (as shown in Supplementary Figures 4a and 4b).

We have modified the Supplementary Figures 4a and 4b to emphasize the smaller overlap region in BiGa:LuIG compared to YIG.

4. BLS results were presented for a small range of wavevectors, and parameters such as exchange constant were obtained. However, more details would be helpful. For example, both phonons and magnons can produce Brillouin peaks in the same frequency and wavevector range. Did they observe phonon peaks? If not, why phonon peaks are absent? If yes, it would be good to present both.

We agree that phonons can be observed by BLS, however our setup is optimized for the observation of magnon response and the phonon response has been purposely suppressed. Therefore our BLS data do not show any phonon peaks.

Instead, we have separately measured the phonon dispersion using the magneto-optical measurement introduced by Y. Hashimoto et al. Nat. Commun. **8**, 15859 (2017) and obtained the values of the phonon velocity for TA and LA phonons as shown in page 4, lines 117-120 of the manuscript.

5. The temperature dependence of the resonant fields was attributed to the change in the exchange constant. How about possible change in phonon dispersion over the same temperature range?

The temperature dependence of the elastic properties of YIG was previously investigated by Y. A. Burenkov et al. Materials. Sci. and Eng. A, **370** 361 (2004), they observed that the phonon velocity decreases as the temperature increases, implying a reduction of the slope of the phonon dispersion with increasing temperatures. Under this scenario, the touching condition between

the magnon and phonon dispersions can be obtained at lower magnetic fields for increasing temperature, contrary to our observation.

This further supports our assumption that the increasing magnetic fields required for the observation of magnon-polaron SSE, can be explained by a softening of the magnon dispersion. In fact, the magnetic field (H_{TA}) increases by a factor of 2.3 between 300 and 380 K, while the phonon velocity remains almost constant ($< 5\%$ decrease over the same temperature range, Fig. 1 of the paper by Y. A. Burenkov et al.), indicating that the effect of variation in the phonon dispersion is very small. We have now added a corresponding discussion about this point in page 8, lines 224-229 of the manuscript.

6. There is no ref. 38 as they quoted in the manuscript.

We apologize for the mistake, we have carefully revised the new version of the manuscript and corrected all typos and mistakes where found.

Reviewer #3 (Remarks to the Author):

This work reports on the resonant enhancement of the spin-Seebeck effect (SSE) due to magnon-phonon coupling at a certain value of magnetic field. The enhancement occurs when a specific mode of the magnon dispersion, which is more or less parabolic, shifts upwards in energy with an external applied magnetic field, and becomes tangential with an acoustic phonon mode, which is linear and field-independent. The hybridization between phonon and magnon modes then covers a wider range of momentum space. The theory for the effect, worked out in Ref. 8, is based on the need for heat to couple from the phonon system to the magnon system. This, according to the authors, is proportional to a factor η defined as the ratio of the phonon lifetime to the magnon lifetime. The paper notices that the resonant enhancement of the SSE is much larger in a Lu-Bi-Ga-Fe garnet (LBGFG) than in the more well-known yttrium-iron garnet (YIG). This, in turn, they attribute to the lower magnon lifetime

in LBGFG, and I agree. The paper further points out that the dispersions of the magnon branches in YIG deviate strongly from parabolic.

Basically, because I think that this work opens the prospect of using SSE in the analysis of hybridized modes, my opinion is that the results are new enough and important enough to be published in Nat. Comm. However, I would like to point a few things out.

We thank the reviewer for the positive comments and recommending publication of the manuscript. We provide a point-by-point response to all comments.

1. The magnitude difference between the result on YIG and on LBGFG could have a contribution that is not intrinsic to the magnon-phonon coupling but simply to the way the data are plotted. The reason for this argument is the difference in thermal conductivity between YIG and LBGFG, which in turn is due to alloy scattering of phonons. Both films are grown on GGG. The heat flux creates a temperature drop across both the GGG and the YIG or LBGFG films. The data are presented in $\mu\text{V}/\text{K}$, where K is the unit for the temperature difference across the structure. But since YIG is more conductive than LBGFG, a larger fraction of the temperature drop across the structure occurs across the film in the LBGFG sample than in the YIG sample. The thermal driving force is therefore higher in that sample, irrespectively of the resonance. Is the effect still present when the SSE effect is plotted as microvolts per unit of heat flux for both samples? This is a quick check the authors should have all data to carry out.

The comparison of the MP peaks between YIG and BiGa:LuIG is performed for the ratio of the magnitude of the peaks normalized by the SSE background signal. Thanks to the comparison of ratios instead of absolute values, we can get rid of any extrinsic effects due to the way the data is plotted.

We apologize if this was not clear in the previous version of the manuscript, we have now included a section titled: "Estimation of the enhancement at the magnon-polaron peaks" in Methods, explaining how the peak magnitude and the relative enhancement with respect to YIG are estimated (this is also briefly mentioned in the caption of Figure 2 of the main text).

2. The factor η in the theory raises questions. I suppose the important parameter is the transfer of energy and momentum between magnons and phonons. Why consider the total lifetime of these quasi-particles? Isn't what matters the fraction of collisions of magnons with phonons to those of magnons with everything else? Secondly, the two terms should be accessible experimentally for YIG from existing neutron data: these lifetimes are given by the bandwidth in energy of the measured magnon and phonon dispersions. Thirdly, if a resonant condition occurs, that too should be visible experimentally via neutron scattering, similarly to Delaire's work on phonons (Nature Materials volume 10, pages 614–619 (2011)).

We thank the reviewer for the comment. The theory of the magnon-polaron SSE considers the magnon-phonon transport in the strongly coupled regime [Refs. 7 and 8 of the manuscript]. Here, at the crossings of the magnon and phonon dispersions hybridized particles are formed, that are neither pure magnon or phonon, but magnon-polarons. In this picture it is not possible to discuss in terms of collisions between magnon and phonons at the crossing of the dispersions, since they are not eigenstates of the Hamiltonian.

We have modified the discussion to improve the description about magnon-polarons and their effect on the SSE (pages: 3-4, lines: 83-90).

We agree with the reviewer that neutron scattering can be used to access the magnon and phonon lifetimes. However, while neutron scattering has been performed in YIG [J. S. Plant, J. Phys. C 10, 4805 (1977), A.J. Princep et al. npj Quantum Materials 2 63 (2017), H. Man et al. PRB 96 100406(R) (2017), S. Shamoto et al. PRB 97 054429 (2018)], information for $\text{Lu}_2\text{BiFe}_4\text{GaO}_{12}$ is lacking. Unfortunately we cannot pursue this experiment, since this is a very limited access experiment, requiring long time frames and large sample sizes.

3. The manuscript spends a lot of effort on calculating the magnon dispersions, which deviate considerably from parabolic. While important to the discussion here, this was known for a long time (Ref. 33 and Plant, J. Phys. C. 10, 4805-4814 (1977)).

We agree that in the case of YIG the magnon dispersion has been previously studied and reported, we have added a comment with corresponding references at the beginning of the section: "Magnon dispersion:..." in page 5, lines 136-139.

4. type p4 line 3, "..., there ARE two possible..."

We thank the reviewer for pointing out the mistake, we have corrected it in the new version.

Reviewers' Comments:

Reviewer #1:

Remarks to the Author:

The authors have attempted to address all the comments raised by the reviewers and improved their manuscript accordingly. I am satisfied with the revised manuscript and recommend it publication.

Reviewer #2:

Remarks to the Author:

The authors have addressed my comments and suggestions. I am ok with accepting it for publication.

Reviewer #3:

Remarks to the Author:

Having reviewed the reply of the authors to the reviewer comments, I think all questions were answered. I think the paper is suitable for publication in its present form.

All my (Rev. 3) comments are addressed satisfactorily, including comment 2, the reply to which I still would like to comment on. I do accept the authors' argument that INS is a measurement that goes beyond the scope of the paper. However, the argument that magnons and phonons, separately, are not eigenstates of the Hamiltonian is not a response to my comment because it is based solely on a theory. INS should see a signal that looks like an avalanche at the (q,E) values where the coupling happens, in effect strengthening the argument for the authors' theoretical model.

Separately, I read through the authors' reply to the other reviewers comments, and find the replies reasonable. However, the reply to Rev. 1, comment 6 raises a new question. The figure 6 shows the temperature-dependence of the SSE in microvolt/K, which shows a anomaly at 175 K, where a spin reorientation was previously reported. What would the data look like if plotted as microvolts of Spin-Seebeck signal divided by the heat flux instead of the temperature gradient? As reported the SSE signal is the convolution of the sample's thermal conductivity and SSE voltage. (I don't need to see this plot, I just suggest that the authors consider it).

REVIEWERS' COMMENTS:

Reviewer #1 (Remarks to the Author):

The authors have attempted to address all the comments raised by the reviewers and improved their manuscript accordingly. I am satisfied with the revised manuscript and recommend it publication.

We thank the reviewer for recommending the manuscript for publication.

Reviewer #2 (Remarks to the Author):

The authors have addressed my comments and suggestions. I am ok with accepting it for publication.

We thank the reviewer for accepting the manuscript for publication.

Reviewer #3 (Remarks to the Author):

Having reviewed the reply of the authors to the reviewer comments, I think all questions were answered. I think the paper is suitable for publication in its present form.

All my (Rev. 3) comments are addressed satisfactorily, including comment 2, the reply to which I still would like to comment on. I do accept the authors' argument that INS is a measurement that goes beyond the scope of the paper. However, the argument that magnons and phonons, separately, are not eigenstates of the Hamiltonian is not a response to my comment because it is based solely on a theory. INS should see a signal that looks like an avalanche at the (q,E) values where the coupling happens, in effect strengthening the argument for the authors' theoretical model.

Separately, I read through the authors' reply to the other reviewers comments, and find the replies reasonable. However, the reply to Rev. 1, comment 6 raises a new question. The figure 6 shows the temperature-dependence of the SSE in microvolt/K, which shows an anomaly at 175 K, where a spin reorientation was previously reported. What would the data look like if plotted as microvolts of Spin-Seebeck signal divided by the heat flux instead of the temperature gradient? As reported the SSE signal is the convolution of the sample's thermal conductivity and SSE voltage. (I don't need to see this plot, I just suggest that the authors consider it).

We thank the reviewer for the acceptance of the manuscript for publication.

We have also take into consideration his/her last comment and in the figure below we show the temperature dependence of the SSE normalized by the heat flux. The data further confirms that there is no SSE anomaly at 175 K for our film.

Figure 1. Magnitude of the SSE at different temperatures obtained after normalizing by the heat flux